# Rescue Therapy with Furazolidone in Patients with at Least Five Eradication Treatment Failures and Multi-Resistant *H. pylori* infection

**DOI:** 10.3390/antibiotics10091028

**Published:** 2021-08-24

**Authors:** Elena Resina, Javier P. Gisbert

**Affiliations:** Centro de Investigación Biomédica en Red de Enfermedades Hepáticas y Digestivas (CIBEREHD), Gastroenterology Unit, Instituto de Investigación Sanitaria Princesa (IIS-IP), Hospital Universitario de La Princesa, Universidad Autónoma de Madrid (UAM), 28049 Madrid, Spain; elenaresina93@gmail.com

**Keywords:** *Helicobacter pylori*, furazolidone, rescue, refractory, resistance

## Abstract

*Helicobacter pylori* infection may persist after multiple eradication treatments. The aim of this study was to evaluate the efficacy and safety of a furazolidone-based rescue regimen in hyper-refractory patients. A unicentre, prospective study was designed. Patients in whom five or more treatments had consecutively failed were included. All patients had previously received bismuth and key antibiotics, such as amoxicillin, clarithromycin, metronidazole, levofloxacin, tetracycline, and rifabutin, and had positive *H. pylori* culture, demonstrating resistance to clarithromycin, metronidazole, and levofloxacin. A quadruple regimen with furazolidone (200 mg), amoxicillin (1 g), bismuth (240 mg), and esomeprazole (40 mg) was prescribed twice a day for 14 days. Eradication was confirmed by the stool antigen test. Compliance was determined through questioning, and adverse effects using a questionnaire. Eight patients (mean age 56 years, 63% men, 38% peptic ulcer disease, 12% gastric cancer precursor lesions, and 50% functional dyspepsia) were included. Per-protocol and intention-to-treat eradication rates were 63%. Compliance was 100%. Adverse effects were reported in two (25%) patients, and all were mild. Even after five or more previous *H. pylori* eradication failures, and a multi-resistant infection, rescue treatment with furazolidone may be effective in approximately two-thirds of the cases, constituting a valid strategy after multiple previous eradication failures with key antibiotics such as clarithromycin, metronidazole, tetracycline, levofloxacin, and rifabutin.

## 1. Introduction

*Helicobacter pylori* (*H. pylori*) is a highly prevalent worldwide infection that is the main cause not only of gastritis, but also of peptic ulcer and gastric cancer [1].

It has been established that not only inflammatory alterations, but also epigenetic changes, especially DNA methylation and histone changes, and interactions between *H. pylori* and human host microbiota, may have a role in the development of these diseases [2,3].

*H. pylori* eradication is indicated in several conditions including peptic ulcer disease, mucosa-associated lymphoid tissue lymphoma, gastric cancer precursor lesions, or iron deficiency anaemia of unknown cause [4,5]. Nevertheless, more recently, consensus has been reached on eradicating *H. pylori* regardless of the associated clinical condition [5,6].

The most effective first-line *H. pylori* eradication treatments for each particular setting must always be prescribed. These treatments should be regimens that have demonstrated achieving cure rates ≥90% in this setting, even when prescribed empirically. The choice of an empirical rescue treatment depends on which treatment is used initially; in the case of *H. pylori* therapy failure, none of the antibiotics against which *H. pylori* has probably become resistant should be administered. Another alternative is to perform antibiotic susceptibility testing and to prescribe a tailored eradication regimen accordingly [7,8].

In some cases, *H. pylori* infection persists even after multiple eradication treatments. High-dose proton pump inhibitor–amoxicillin dual regimens or rifabutin-based therapies have generally been recommended for patients with several previous eradication failures [9,10], but these regimens also fail in some cases. It should be noted that these hyper-refractory cases are an extremely rare exception, with less than 1% of patients remaining infected with *H. pylori* after multiple (>4–5) eradication treatments [8]. However, these hyper-refractory patients pose a challenge since the therapeutic options in these cases are very limited.

Furazolidone is a synthetic nitrofuran, a monoamine oxidase inhibitor with a known broad spectrum of antimicrobial activities, active against Gram-positive and Gram-negative bacteria as well as various protozoa. It has been used in both humans and animals [11] and empirically used for over 20 years to treat peptic ulcer disease successfully [12].

In monotherapy, furazolidone has shown high antimicrobial activity against *H. pylori*, and most first-line treatments, including this antibiotic, have achieved eradication rates of more than 80% [13,14,15]. The rate of furazolidone primary resistance is very low, and the risk of developing secondary resistance has been rarely reported, as with bismuth or amoxicillin [16]. In this regard, furazolidone has shown to be effective in patients in whom other eradication treatments previously failed [17].

Furazolidone has been used when other regimens, including clarithromycin or metronidazole, failed [18,19] or where primary resistance to clarithromycin and levofloxacin are so high that it makes these treatments unadvisable [20,21]. In fact, the Fifth Chinese National Consensus Report recommended furazolidone, amoxicillin, bismuth, and proton pump inhibitor quadruple therapy as one of the first-line regimens for *H. pylori* therapy, since estimated resistance to clarithromycin and metronidazole in that country exceeds 20% and 40%, respectively [22].

Therefore, the aim of the present study was to evaluate the efficacy of a furazolidone rescue regimen in patients with at least five eradication failures and multi-resistant *H. pylori* infection.

## 2. Results

### 2.1. Demographic Variables

From June 2018 to February 2021, we identified fifteen patients who had at least five previous *H. pylori* eradication failures (with a mean of six failed treatments per patient), who are therefore potentially eligible for furazolidone-based treatment. Three patients declined treatment because they refused to undergo upper gastrointestinal endoscopy or further antibiotic treatment. Four patients underwent upper gastrointestinal endoscopy and had biopsies taken but they had negative culture results, thus they could not receive furazolidone. Finally, eight patients had a positive culture for *H. pylori* demonstrating simultaneous resistance to at least clarithromycin, metronidazole, and levofloxacin. Patients had previous diagnosis of peptic ulcer disease (38%), gastric cancer precursor lesions (12%), relatives with gastric cancer (12%), or functional dyspepsia (50%).

The mean age ± standard deviation of the eight patients that received furazolidone was 56 ± 7 years, and 63% were male.

The baseline characteristics of the patients that received furazolidone, and their previous—failed—eradication treatments, are detailed in Table 1.

### 2.2. Compliance with the Protocol and Loss to Follow-Up

All patients returned for their follow-up visit (no patient was lost to follow-up). All patients complied with the protocol (that is, reported that they took 100% of the prescribed medication).

### 2.3. Tolerance to Eradication Therapy

Adverse effects were reported in two patients (25%). One experienced nausea, and another had abdominal discomfort. All the adverse effects were classified as mild, and symptoms were only present while the patient was on medication.

### 2.4. Efficacy of Eradication Therapy

Eradication was achieved in 63% (5/8; 95% confidence interval, 25–92%) of patients, both by per-protocol and intention-to-treat analysis.

## 3. Discussion

Nowadays, in addition to having a good understanding of first-line eradication regimens, gastroenterologists must also be prepared to face *H. pylori* treatment failures and even hyper-refractory cases.

Furthermore, *H. pylori* infection has a powerful impact on both the stomach habitat, the immune status, and the epigenetic changes of the host. Understanding of the mechanisms involved in these processes will lead us to an improved management of the infection and provide future treatment options [2,23].

Antibiotic resistance is the major factor affecting our ability to eradicate *H. pylori*. It is reaching alarming levels worldwide, with a post-treatment multidrug resistance rate (resistance to ≥3 antibiotics of different classes) of >23–36%, as informed in a recent review [24]. In Spain, *H. pylori* triple resistance (clarithromycin, metronidazole, and levofloxacin) reached approximately 2% [25].

The European Registry on the Management of *H. pylori* (Hp-EuReg), which analyses more than 3000 patients with culture-positive *H. pylori* infection, recently reported 21% resistance to single clarithromycin and 11% dual resistance (clarithromycin and metronidazole) in untreated patients. Antibiotic resistance increased markedly from the first treatment, reaching more than 37% dual resistance in second-line treatment, and approximately 30% triple resistance in patients with at least three treatment failures [26]. These findings confirm the unusual prevalence of *H. pylori* triple resistance in our setting, which may explain the reduced sample size included in the present study.

After failure of a third-line treatment, a fourth-line with rifabutin is generally recommended [7]. However, in the largest multi-centre study including this antibiotic, an eradication rate of 52% was reported [27], although a recent review reported an eradication rate of 73% [9]. All our patients had previously received (and failed) rifabutin treatment.

High-dose proton pump inhibitor–amoxicillin dual therapy (±bismuth) is another frequently used rescue treatment. Two recent systematic reviews concluded that this therapy was equivalent to generally recommended first-line or rescue regimens, such as triple therapy, bismuth quadruple therapy, and non-bismuth quadruple therapy, with fewer adverse effects [10,28]. In the present study, 88% (7/8) of the patients were previously treated with high-dose proton pump inhibitor–amoxicillin dual therapy. The treatment failed in all of them despite the low reported rate of amoxicillin resistance [10], highlighting the hyper-refractoriness of our patients.

Our results with the combination of furazolidone, amoxicillin, bismuth, and a proton pump inhibitor for 14 days are encouraging, with an eradication rate superior to 60%. It must be highlighted that this rescue regimen was prescribed in all the cases after at least five eradication failures with key antibiotics such as clarithromycin, metronidazole, tetracycline, levofloxacin, and rifabutin.

Regarding the eradication assessment method, the accuracy of the monoclonal stool antigen test has been demonstrated in several systematic reviews. It is accurate both for the initial diagnosis of *H. pylori* infection and for confirmation of eradication after treatment, being comparable to other non-invasive tests such as the urea breath test [29,30]. It is also inexpensive, easy to perform, and comfortable for the patient.

Several studies have analyzed the outcomes of furazolidone-containing regimens as first-line therapy. A pooled-data analysis from Zullo et al. concluded that, following furazolidone-based first-line therapy, *H. pylori* eradication rates achieved 75–80% [11]. More encouraging results were reported in other studies, showing eradication rates for furazolidone and bismuth containing quadruple therapy, of 93% [31] and 95% [15,17].

Similarly to the present study, several groups studied the utility of furazolidone-based treatment as a rescue therapy. Cheng et al. analyzed data from 60 patients after one or more treatment failures. All patients received 10 mg of rabeprazole, 1 g of amoxicillin, 100 mg of furazolidone, and 220 mg of bismuth subcitrate twice daily for 14 days. The *H. pylori* eradication rate was 89% [14]. Other studies used a regimen similar to ours, a furazolidone- and amoxicillin-based quadruple therapy as rescue therapy, but only 13% of the patients had one prior treatment, and only 2% had ≥2 prior treatments, reporting an eradication rate of 91% [15].

In a study by Song et al., 584 patients were treated with furazolidone-containing quadruple regimens [17]. Noteworthy, only 0.5% of patients in this study had two or more treatment failures. Eradication rate decreased from 95% as first-line treatment to 87% when furazolidone was used as rescue therapy. Some authors compared four bismuth-containing quadruple therapies, two of them containing furazolidone, in patients who did not respond to previous treatment. All therapies achieved an eradication rate greater than 90%, being 95–99% for furazolidone- and amoxicillin- based therapy [21].

A previous review of furazolidone-based therapies found that, following one to three previous treatments, the infection was cured in approximately 80% patients [11], and, as a third-line therapy, some authors reported a 65% eradication rate [32]. In our study, we found an eradication rate of 63% in patients with at least five eradication treatment failures (and harboring *H. pylori* strains resistant to clarithromycin, metronidazole, and levofloxacin). Thus, to the best of our knowledge, our study is the first one to evaluate the furazolidone-based regimen in such hyper-refractory patients and multi-resistant infection, achieving encouraging results.

Regarding therapy duration, a previous review analyzed six different furazolidone-based antibiotic combinations. The comparison between 7- and 14-day regimens found a similar efficacy by intention-to-treat analysis (75.5% vs. 75.9%), while a significantly higher eradication rate was achieved with the prolonged regimen in per-protocol analysis (83.5% vs. 82.3%; *p* < 0.05) [11]. Other studies compared 7- and 14-day regimens of furazolidone-based quadruple therapy, finding an eradication rate of 89% with the prolonged regimen which decreased to 82% with a one-week treatment [14]. Other authors reported eradication rates for 10-day and 14-day regimens of 93.7% and 98.2%, respectively [17]. According to the previous experience, we decided to prescribe a 14-day regimen in our study, mainly taking into account the hype-refractory nature of our patients.

Concerning the furazolidone dose, a significantly higher cure rate was found with high doses (200 mg b.i.d.) compared to low doses (100 mg b.i.d.) (84.9% vs. 75.3%; *p* < 0.001) [11]. Similar results were found in a recent meta-analysis, confirming that a higher daily dosage of furazolidone exhibited a better eradication rate than the lower daily dosage (83.5% vs. 70.7%, for the intention-to-treat analysis) [31]. The aforementioned study by Cheng et al. showed an eradication rate of 89% using a quadruple furazolidone treatment (100 mg b.i.d.) for 14 days. The eradication rate increased to 90% when a higher dose of furazolidone was administered (100 mg t.i.d.); a regimen similar to that was also used in our study [14].

The rate of adverse effects of furazolidone treatment in recent *H. pylori* studies ranged from 8% to 33% [11,17]. In our study, the incidence was 25%, in accordance with previous data. Zullo et al. reported an overall side effect incidence of 33%, with only 3.8% of them considered severe [11]. Another study including more than 900 patients found that 17% of patients experienced one or more treatment-related adverse effects, with only 2.8% classified as severe, with nausea, abdominal pain, and dizziness being the most frequent [15]. The meta-analysis performed by Zhuge et al. found no statistically significant differences in adverse effect rates compared to other furazolidone-free regimens [31]. Finally, other authors found furazolidone to be even safer than commonly used drugs such as metronidazole [33]. In our study, no severe side effects were reported, and the most frequent ones affected the gastrointestinal tract (nausea and abdominal discomfort), in agreement with previous studies [11,15,17,31,33]. Although some authors warn of the risk of carcinogenesis with the use of furazolidone [11], large studies and meta-analyses have not proven this effect and, therefore, it is considered a safe agent [33].

In summary, the present study shows that, even after five or more previous *H. pylori* eradication failures and in the presence of multi-resistant infection, i.e., an extremely challenging setting, a rescue treatment with furazolidone may be effective in two-thirds of the cases. Therefore, furazolidone-based rescue therapy constitutes a valid, safe, and affordable strategy, and one of the few options after multiple previous eradication failures with key antibiotics, such as clarithromycin, metronidazole, tetracycline, levofloxacin, and rifabutin.

## 4. Materials and Methods

### 4.1. Patients

We prospectively studied consecutive patients with *H. pylori* infection, in whom at least five treatments had consecutively failed to eradicate *H. pylori* infection. *H. pylori* eradication failure with the last treatment was defined by a positive ^13^C-urea breath test 4–8 weeks after completion of treatment. The exclusion criteria were as follows: (i) age under 18 years; (ii) presence of clinically significant associated conditions (neoplastic diseases, coagulation disorders, and hepatic, cardiorespiratory, or renal diseases; (iii) previous gastric surgery; and (iv) allergy to any of the drugs used in the study. All patients were offered to undergo upper gastrointestinal endoscopy for culture sampling and to test for multiple resistance to the antibiotics commonly used in eradication treatment (at least to clarithromycin, metronidazole, and levofloxacin), since this was a requirement of the Spanish Agency for Medicines and Health Products (AEMPS) for furazolidone treatment.

### 4.2. Therapy

A quadruple eradication regimen with furazolidone (200 mg b.i.d.), amoxicillin (1 g b.i.d.), bismuth (240 mg b.i.d.), and esomeprazole (40 mg b.i.d.) was prescribed for 14 days. All drugs were administered together after breakfast and dinner. The prescription of furazolidone was approved by AEMPS, the pharmacy service, and the hospital’s medical direction, and informed consent was obtained from all the patients. Patients were informed of potential side effects (mainly nausea and dizziness) during the treatment period. Compliance with therapy was defined as intake of 100% of the medication prescribed, and was determined by a questionnaire (applied by the gastroenterologist responsible for treating the patient). The incidence of adverse effects was evaluated using a specific questionnaire (Appendix A) [34].

Adverse effects were classified as mild, moderate, or severe, depending on their intensity.

### 4.3. Diagnostic Methods to Confirm H. pylori Eradication

Eradication of *H. pylori* was defined by a negative stool antigen test, according to standard clinical practice in our center, performed 4–8 weeks after the end of retreatment. The test used was “LIAISON^®^ Meridian *H. pylori* SA”, a chemiluminescent immunoassay (CLIA) stool antigen test, which applies monoclonal antibodies (DiaSorin Iberia S.A., Madrid, Spain) [35].

### 4.4. Statistical Analysis

Categorical variables are expressed as percentages, 95% confidence intervals, and quantitative variables as mean ± standard deviation. Efficacy of *H. pylori* eradication was assessed by intention-to-treat analysis (including all eligible patients enrolled in the study regardless of compliance with the study protocol; patients with data unsuitable for evaluation were assumed to have been unsuccessfully treated) and by per-protocol analysis (excluding patients whose compliance with therapy was poor and patients with data unsuitable for evaluation after therapy).

## 5. Conclusions

Even after five or more previous *H. pylori* eradication failures, and a multi-resistant infection, rescue treatment with furazolidone may be effective in approximately two-thirds of the cases, constituting a valid strategy after multiple previous eradication failures with key antibiotics such as clarithromycin, metronidazole, tetracycline, levofloxacin, and rifabutin.

## Figures and Tables

**Table 1 antibiotics-10-01028-t001:** Baseline characteristics of patients treated with furazolidone, and their previous failed eradication treatments.

Patient	Gender	Age (Years)	Diagnosis	Number of Previous Failed Treatments	Previous Treatments							Eradication Assessment Method	Eradication Success with Furazolidone
					1	2	3	4	5	6	7		
1	Female	59	PUD	6	PPI + C + A	PPI + A + L	PPI + B + T + M	PPI + C + A + M	PPI + A + R	PPI + A high dose + B		SAT	Yes
2	Female	48	PUD	5	PPI + C + A	PPI + A + L	PPI + B + T + M	PPI + A + R	PPI + A high dose + B			SAT	Yes
3	Male	69	GCPL	6	PPI + C + A	PPI + B + T + M	PPI + A + L	PPI + A + R	PPI + C + A + M	PPI + A high dose + B		SAT	Yes
4	Male	57	FD	7	PPI + C + A	PPI + A + L	PPI + B + T + M	PPI + C + A + M	PPI + A + R + B	PPI + P	PPI + A high dose + B	SAT	Yes
5	Male	47	FD + RGC	6	PPI + C + A	PPI + C + L	PPI + A + L + B	PPI + C + A + M	PPI + B + T + M	PPI + A + R		SAT	Yes
6	Male	52	PUD	6	PPI + C + A	PPI + C + M	PPI + A + L	PPI + B + T + M	PPI + A + R + B	PPI + A high dose + B		SAT	No
7	Male	56	FD	7	PPI + C + A	PPI + A + L	PP I + A + M	PPI + B + T + M	PPI + A + R + B	PPI + A high dose + B	PPI + C + A + M + B	SAT	No
8	Female	56	FD	5	PPI + C + A + M	PPI + P	PPI + A + L + B	PPI + A + R	PPI + A high dose + B			SAT	No

SAT: Stool antigen test, UD/FD: uninvestigated dyspepsia/functional dyspepsia, PUD: peptic ulcer disease, RGC: Relative with gastric cancer, GCPL: Gastric cancer precursor lesions, PPI: proton pump inhibitor, C: Clarithromycin, A: Amoxicillin, M: Metronidazole, L: Levofloxacin, R: Rifabutin, B: Bismuth, T: Tetracycline, P: Pylera (three-in-one single capsule, bismuth-containing quadruple therapy).

## Data Availability

The data presented in this study are available on request from the corresponding author.

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
