# Peer review of "Rescue Therapy with Furazolidone in Patients with at Least Five Eradication Treatment Failures and Multi-Resistant H. pylori infection"

_antibiotics, 2021, doi:10.3390/antibiotics10091028_

Round 1
Reviewer 1 Report
In this article, Elena Resina and Javier P. Gisbert reported the a unicentre, prospective study that describe the efficacy of the furazolidone based quadruple therapy as a rescue therapy.
This study well describe the efficacy of this regimen and quite interesting except several points.
1. The author used the stool antigen test, however, this test is not very common to confirm the H. pylori eradication except pediatric patients.
Please explain the reason and add some reference about this test.
2. There is safety issue regarding the furazoline usage for the H. pylori eradication because Federal Register stating that the company has voluntarily requested withdrawal of approval of these applications. Please explain the safety issues more clearly for the wide usage of this antibiotics in the discussion section.
3. If there is recent reference that study the in vitro culture data for the multidrug resistant H. pylori. this article could be more interesting to the readers.
Author Response
Dear reviewer,
After a careful revision of our article proposal, based on the suggestions made by the reviewers, we have proceeded to send it for a new evaluation. In the new manuscript, we have highlighted in red the modifications made to the original text.
We would like to express our sincere gratitude to the reviewers for their valuable work; their annotations have allowed us not only to significantly improve the manuscript but also to reflect on future research.
In the following of this letter, we detail how we have addressed the reviewers' suggestions in the new version of our article proposal. We hope that the work we have done will meet with the final approval of the Editorial Team. Should this not be the case, all authors are at your disposal to resolve any issues or to proceed with further revisions to the extent necessary.
- The author used the stool antigen test; however, this test is not very common to confirm the H. pylori eradication except pediatric patients. Please explain the reason and add some reference about this test.
- As demonstrated in the article entitled: “Opekun, A.R.; Zierold, C.; Rode, A.; Blocki, F.A.; Fiorini, G.; Saracino, I.M.; Vaira, D.; Sutton, F.M. Clinical Performance of the Automated LIAISON® Meridian H. pylori SA Stool Antigen Test. BioMed research international 2020, 2020, 7189519, doi:10.1155/2020/7189519” that is included in our article in the reference number 35, it is proved the good accuracy of the stool antigen test that we used in our hospital, the study includes 277 patients (100% agreement with histopathology as a gold standard) and highlights the excellent accuracy of the test; sensibility 95,5 % and specificity 97,6%.
- In addition, several systematic reviews demonstrate high diagnostic accuracy in both naive and post-treatment patients. We have now included these references in the article:
- Gisbert JP, de la Morena F, Abraira V. Accuracy of monoclonal stool antigen test for the diagnosis of H. pylori infection: a systematic review and meta-analysis. Am J Gastroenterol. 2006 Aug;101(8):1921-30. doi: 10.1111/j.1572-0241.2006.00668. x. Epub 2006 Jun 16. Erratum in: Am J Gastroenterol. 2006 Oct;101(10):2445. PMID: 16780557. Including twelve studies, 957 patients, assessed the monoclonal SAT to confirm eradication after therapy. Pooled sensitivity, specificity, LR+, and LR- were 0.93, 0.96, 17, and 0.1.
- Stool Antigen Tests for Helicobacter pyloriInfection: A Review of Clinical and Cost-Effectiveness and Guidelines [Internet]. Ottawa (ON): Canadian Agency for Drugs and Technologies in Health; 2015 Jan 8. PMID: 25632493. Sensitivity of monoclonal SAT ranged 90.0% to 92.4%, and specificity ranged from 91.0% to 100%.
- It is also an inexpensive technique, easy to perform and comfortable for the patient.
- There is safety issue regarding the furazoline usage for the H. pylori eradication because Federal Register stating that the company has voluntarily requested withdrawal of approval of these applications. Please explain the safety issues more clearly for the wide usage of this antibiotics in the discussion section.
- We included recent systematics reviews with a large number of patients that describe the rate and severity of the adverse effects.
- Zullo et al 2012: suggest that patients have to be clearly informed about the possible genotoxic and carcinogenetic effects for which furazolidone use is not approved in developed countries. No other study mentions these genotoxic or carcinogenic effects since at the doses used in real practice for H. pylori treatment this effect is negligible and in general all reviews with large numbers of patients consider the treatment to be safe. Furthermore, our study does not propose furazolidone for generalised use as a first line treatment but as a rescue treatment in hyper-refractory patients. In any case, all patients were informed of the possible adverse effects before the start of treatment.
- Graham et al 2012: Agency for Research on Cancer (IARC) categorizes agents in categories or groups, Furazolidone is a category 3 agent: Unclassifiable as to carcinogenicity in humans. The IARC report states: “Furazolidone has been produced commercially since 1955. It is used in human and veterinary medicine as an antibacterial and antiprotozoal agent. No data were available to assess the teratogenicity or chromosomal effects of this compound in humans. No case report or epidemiological study of the carcinogenicity of furazolidone was available to the Working Group. Evaluation: No evaluation of the carcinogenicity of furazolidone to experimental animals could be made. In the absence of epidemiological data, no evaluation of the carcinogenicity of furazolidone to humans could be made.” Furazolidone is no longer marketed in the United States because the Shire company decided the market was too small and informed the FDA that they would no longer market it. The FDA responded that they would “publish a notice in the Federal Register stating that (the company has) voluntarily requested withdrawal of approval of these applications because (it has) stopped marketing the drug products under the NDA”.
- Zhugue et al. 2018. 18 studies included: No differences in the incidence of adverse effects between Furazolidone based therapy and control group. Only fever, weakness, and anorexia, was higher in the furazolidone group than in the control group (RR 3.99).
- Zhang et al 2018. Big Serie with 992 patients. 16.8% experienced one or more treatment-related adverse events (Table 3). The common adverse events including abdominal pain in 39 (4.5%), nausea in 20 (2.3%), dizziness in 11 (1.3%), fatigue in 11 (1.3%), anorexia in 13 (1.5%), and skin rash/pruritus in 18 (2.1%) were reported. Twenty-four (2.8%) patients experienced severe treatment-associated adverse events necessitating premature discontinuation, being the most frequent skin rash.
- If there is recent reference that study the in vitroculture data for the multidrug resistant H. pylori. this article could be more interesting to the readers.
- We added a new recent citation of The European Registry on the Management of pylori (Hp-EuReg) which analyses more than 3,000 patients with culture-positive H. pylori, recently reported 21% resistance to single clarithromycin and 11% dual resistance (clarithromycin and metronidazole) in untreated patients. Antibiotic resistance increased markedly from the first treatment, reaching more than 37% dual resistance in second-line treatment, and reaching approximately 30% triple resistance in patients with at least 3 treatment failure. Reference: Bujanda L, Nyssen OP, Cosme A, Bordin DS, Tepeš B, Perez-Aisa A , Vaira D, Caldas M, Castro-Fernandez M, Lerang F, Leja M, Rodrigo L, Rokkas T, Kupcinskas L, Perez-Lasala J, Jonaitis L, Shvets O, Gasbarrini A, Simsek H, Axon ATR, Buzas GM, Machado JC, Niv Y, Boyanova L, Goldis A, Lamy V, Tonkic , Marlicz W, Beglinger C, Venerito M, Bytzer P, Capelle LG, Milosavljevic T, Veijola L, Molina Infante J, Vologzhanina L, Fadeenko G, Ariño I, Fiorini G, Resina E, Muñoz R, Puig I, Megraud F, O’Morain C., Gisbert JP, On behalf of the Hp-EuReg Investigators. Helicobacter pyloriantibiotic resistance: data from the european registry on H. pylori management (Hp-EuReg). United European Gastroenterology Journal 2020, 8(8S) 257.
Yours sincerely,
The authors.

Reviewer 2 Report
The manuscript entitled “Rescue therapy with furazolidone in patients with at least five eradication treatment failures and multi-re- 3 sistant H. pylori infection” is well structured and very interesting, the supporting literature is adequate.
However, I have some small suggestions for the authors:
- pylori infections are common, as are antibiotic resistance when the pathogen is not adequately adapted.
However, changes in H. pylori infections are known to cause epigenetic changes; therefore genetic tests should also be performed in a study population under antibiotic treatment.
In light of this, I invite the authors to the evaluation to introduce these studies (doi: 10.3390 / biom9060237. PMID: 31216758; doi: 10.1371 / journal.pone.0156671; doi: 10.1371 / journal.pone.0222295) where it is also possible alternative approaches in the treatment of H. pylori infections.
Author Response
Dear reviewer,
After a careful revision of our article proposal, based on the suggestions made by the reviewers, we have proceeded to send it for a new evaluation. In the new manuscript, we have highlighted in red the modifications made to the original text.
We would like to express our sincere gratitude to the reviewers for their valuable work; their annotations have allowed us not only to significantly improve the manuscript but also to reflect on future research.
In the following of this letter, we detail how we have addressed the reviewers' suggestions in the new version of our article proposal. We hope that the work we have done will meet with the final approval of the Editorial Team. Should this not be the case, all authors are at your disposal to resolve any issues or to proceed with further revisions to the extent necessary.
- However, changes in H. pylori infections are known to cause epigenetic changes; therefore genetic tests should also be performed in a study population under antibiotic treatment. In light of this, I invite the authors to the evaluation to introduce these studies (doi: 10.3390 / biom9060237. PMID: 31216758; doi: 10.1371 / journal.pone.0156671; doi: 10.1371 / journal.pone.0222295) where it is also possible alternative approaches in the treatment of H. pylori infections.
- Thank you very much for your comment. We have found these articles very interesting; we agree that they can complete and contribute with relevant information to our study. We have therefore decided to include and cite them in the introduction and discussion of the manuscript.
Yours sincerely,
The authors.

Reviewer 3 Report
The aim of this study was to evaluate the efficacy and safety of a furazolidone-based rescue regimen in hyper-refractory patients.
This in an interesting study. I have only one minor comment:
- Authors stated that” The incidence of adverse effects was evaluated using a specific questionnaire”. This questionnaire should be presented in the manuscript (even as supplemental material).
Author Response
Dear reviewer,
After a careful revision of our article proposal, based on the suggestions made by the reviewers, we have proceeded to send it for a new evaluation. In the new manuscript, we have highlighted in red the modifications made to the original text.
We would like to express our sincere gratitude to the reviewers for their valuable work; their annotations have allowed us not only to significantly improve the manuscript but also to reflect on future research.
In the following of this letter, we detail how we have addressed the reviewers' suggestions in the new version of our article proposal. We hope that the work we have done will meet with the final approval of the Editorial Team. Should this not be the case, all authors are at your disposal to resolve any issues or to proceed with further revisions to the extent necessary.
- Authors stated that” The incidence of adverse effects was evaluated using a specific questionnaire”. This questionnaire should be presented in the manuscript (even as supplemental material).
- Please find attached the document we used as a specific questionnaire to assess the occurrence of adverse effects. We have now included it as supplementary material (Supplementary Figure S1).
Yours sincerely,
The authors.

Round 2
Reviewer 1 Report
Thank for the authors because they revised the article perfectly.
I have nothing to comment.